# Influence of Intrauterine Inflammation, Delivery, and Postnatal Feeding on the Temporal Changes of Serum Alpha 1 Acid Glycoprotein Levels in Extremely-Low-Birth-Weight Infants

**DOI:** 10.3390/nu14235162

**Published:** 2022-12-04

**Authors:** Yasuhisa Nakamura, Sachiko Iwata, Kyoko Yokoi, Yuko Mizutani, Masatoshi Yoshikane, Koya Kawase, Takenori Kato, Satoru Kobayashi, Haruo Goto, Shinji Saitoh, Osuke Iwata

**Affiliations:** 1Center for Human Development and Family Science, Department of Pediatrics and Neonatology, Nagoya City University Graduate School of Medical Sciences, 1, Kawasumi Mizuho-cho, Mizuho-ku, Nagoya 467-8601, Japan; 2Department of Pediatrics, Nagoya City University West Medical Center, 1-1-1, Hirate-cho, Kita-ku, Nagoya 462-8508, Japan

**Keywords:** alpha 1 acid glycoprotein, delivery, enteral nutrition, extremely-low-birth-weight infant, gut microbiota, infection, inflammation, preterm birth, probiotics, sensitization

## Abstract

Infection remains the primary cause of death in extremely-low-birth-weight infants (ELBWIs). Alpha 1 acid glycoprotein (α1AG), an acute-phase protein, has been shown to be elevated in sporadic cases of septic ELBWIs prior to abnormal clinical signs. To delineate the roles of inflammation, delivery, and feeding in postnatal α1AG changes in ELBWIs, 75 ELBWIs of 26.5 ± 2.2 weeks of gestation born between May 2011 and August 2017 were retrospectively studied. The dependence of α1AG levels obtained on days 0–5 on the clinical variables was examined by incorporating interactions with age, followed by estimations of regression coefficients between clinical variables and α1AG levels at the early and late postnatal ages, defined by their standard deviation. Chorioamnionitis (*p* < 0.001), funisitis (*p* = 0.045), vaginal delivery (*p* = 0.025), enteral feeding (*p* = 0.022), and probiotics (*p* = 0.005) were associated with early α1AG elevations. Hypertensive disorder of pregnancy (*p* < 0.001) and gestational age (*p* = 0.001) were associated with late α1AG elevation; premature rupture of membranes (*p* < 0.001), funisitis (*p* = 0.021), body weight z-scores (*p* < 0.001), and enteral feeding (*p* = 0.045) were associated with late α1AG reduction. Postnatal α1AG changes in ELBWIs were associated with variables representative of age, growth, delivery, inflammation, and enteral feeding, potentially reflecting the process of sensitization to extrinsic microbes in utero, at birth, and thereafter.

## 1. Introduction

Neonatal infection, one of the primary causes of neonatal morbidities, is responsible for the death of over 0.3 million newborns worldwide per annum [1]. Preterm infants are particularly vulnerable to infection. Septicemia is up to 1000-fold more common in preterm infants compared to term infants [2]. Subsequently, approximately one in two deaths of extremely-low-birth-weight infants (ELBWIs) is caused by neonatal infection [3]. A range of biomarkers has been proposed to identify the early signs of neonatal infection. C-reactive protein (CRP) is an established serum biomarker of bacterial infections in newborn infants, children, and adults [4]. However, approximately 35–65% of hospitalized newborn infants are CRP-negative when the first clinical symptom suggestive of a serious infection is noted [5]. Additionally, the elevation of CRP in newborn infants is not specific to infection, as observed following maternal fever, birth asphyxia, and meconium aspiration [6]. Procalcitonin is increasingly used as an early biomarker of infection because of its relatively shorter latent period after the onset of infection and its relatively more specific elevation to bacterial infections in children and adults [7]. However, in preterm infants, procalcitonin levels are high, regardless of the presence of infection during the first few days of life [8], rendering this difficult to use as an early biomarker of neonatal infection following preterm birth.

Alpha 1 acid glycoprotein (α1AG) is an inflammatory reaction protein generated in the liver [9]. Goto et al. proposed a screening algorithm for neonatal infections using α1AG, CRP, and haptoglobin to give a 0–3 composite score of the Acute Phase Reaction (APR) score, which has been implemented as an early screening tool for neonatal infection, mainly in Japanese neonatal intensive care units [10]. Goto et al. further reported sporadic cases of preterm infants whose α1AG levels were elevated prior to changes in CRP and haptoglobin and clinical symptoms of severe systemic infection, which was explained as a consequence of dysbiosis, where the balance of microbiota, such as the gut and immature skin, is deranged [11]. If the α1AG levels temporally mirror the stages of bacterial infection, such as the sensitization to microbes, the breach of the biological barrier, and bacterial colonization within organs, the detection and treatment of systemic infection in preterm infants might be significantly improved. A detailed elucidation of the temporal changes in the α1AG levels related to intrauterine exposure to inflammation, sensitization to environmental microbiota after birth, and the development of systemic infection may help clarify its clinical value as a very early infection biomarker and further reveal the covert process and biological response to bacterial infection during the latent period. However, independent variables regulating the production of α1AG in preterm infants have not been fully identified. More importantly, the serum α1AG levels increase with gestational and postnatal age [12,13]. Therefore, it is crucial to understand the impact of the independent variables of α1AG by incorporating their interactions with age.

We conducted a retrospective observational study in a single-center cohort of ELBWIs to delineate the role of antenatal inflammation, delivery, and postnatal feeding on the temporal changes in the α1AG levels after birth.

## 2. Materials and Methods

The protocol for this retrospective observational study was approved by the Institutional Ethics Review Board of Nagoya City University West Medical Center, Nagoya, Japan (reference number: 21-04-393-45). The requirement for informed parental consent was waived by the review board because this analysis used anonymous clinical data.

### 2.1. Study Population

Between May 2011 and August 2017, a total of 3351 newborn infants were admitted to the neonatal intensive care unit of Nagoya City University West Medical Center. Of these, 78 infants weighing less than 1000 g were retrospectively enrolled in this study. Three infants who died within the first 24 h of life were excluded from this study due to the lack of α1AG data after 24 h of birth, leaving 75 infants available for analysis.

### 2.2. Data Collection

The patients’ clinical data were collected from the electronic medical records, including maternal variables (hypertensive disorders of pregnancy (HDP), premature rupture of membranes (PROM) ≥ 24 h, antenatal steroids and antibiotics, chorioamnionitis, and funisitis), variables at birth (gestational age, mode of delivery, birth weight and its z-score, sex, and 1- and 5-minute Apgar scores), and the clinical events and therapeutic options during hospitalization (blood test findings, blood-culture-positive septicemia, antibiotic use, including prophylaxis, age at the commencement of enteral nutrition and probiotics, intestinal perforation, severe intraventricular hemorrhage, and surgical ligation of ductus arteriosus). HDP was diagnosed using the criteria of the Japan Society for the Study of Hypertension in Pregnancy [14]. In this unit, histopathological examinations of the placenta and umbilical cord are routinely performed by an experienced pathologist for all preterm laborers. Chorioamnionitis was defined as an infiltration of neutrophils identified in the chorionic membranes according to Blanc’s classification (stage II), whereas funisitis was defined as the infiltration of neutrophils within the walls of umbilical vessels or in Wharton’s jelly [15,16]. The birth weight z-scores were calculated using the database for birth weight published by the Japan Pediatric Society as a reference [17]. During the study period, prophylactic antibiotics were only prescribed sparingly when clinical information suggested the presence of a serious systemic infection in the newborn. Severe intraventricular hemorrhage was defined as grade III/IV according to Papile’s classification [18].

### 2.3. Laboratory Studies

In our unit, blood gas, glucose, electrocytes, and serum inflammatory markers were routinely examined every day between day 0 (on admission) and day 5 for ELBWIs. The blood tests between days 2 and 5 were skipped or abbreviated on Sundays/holidays and when the infant’s clinical condition was stable, whereas additional blood tests were performed with clinical signs suggestive of serious infection. For the inflammatory markers, the serum levels of CRP, α1AG, and haptoglobin were measured using a turbidimetric immunoassay (Quick Turbo, Shino-Test Corporation, Tokyo, Japan). The assay limits of detection were 0.3 mg/dL, 20 mg/dL, and 13 mg/dL for CRP, α1AG, and haptoglobin, respectively. All α1AG data from both the routine and extra blood tests performed on days 0–5 were included in the analysis.

### 2.4. Data Analysis

Values are shown as the mean (standard deviation), unless otherwise stated. First, the relationships between the clinical variables were examined to highlight the potential biases derived from their interactions using the Spearman rank correlation coefficient (findings presented in Online Appendix A only). The dependence of α1AG levels on the clinical variables was assessed using generalized estimating equations with a linear model to account for repeated blood sampling within the same infant on different days. The crude dependence of the α1AG levels on the clinical variables was initially assessed without correcting for postnatal age. Then, to highlight the relationships between potential independent variables and temporal changes in the α1AG levels, linear interactions between postnatal age and clinical independent variables were incorporated into the univariate model to predict the α1AG levels. For independent variables with significant interactions with postnatal age, a post hoc simple slope analysis was performed. This analysis estimated the slope of regression lines for the relationships between the independent variables and α1AG levels at two representative time points of “early” and “late” postnatal ages (one standard deviation below and above the mean postnatal age, respectively) [19]. For technical reasons, numeric independent variables were mean-centered in advance. Multivariable models to explain the α1AG levels were not developed because these were not the focus of our study. The statistical findings were presented without correction for multiple comparisons because of the exploratory nature of this study [20]; findings with *p*-values less than but close to 0.05 were interpreted cautiously because of their susceptibility to type-1 errors. All statistical analyses were performed using SPSS Statistics ver. 28 (IBM Corp, Armonk, NY, USA).

## 3. Results

### 3.1. Characteristics of the Study Patients

The final study population comprised 75 infants who were 26.5 ± 2.2 weeks of gestation and 746 ± 162 g at birth (Table 1). Of these, HDP and PROM were observed in 24.0% and 34.7% of the cases, respectively, whereas 80.0% of the infants were delivered via cesarean section. Histopathological chorioamnionitis and funisitis were noted in 18.7% and 17.3% of the infants, respectively. Only 2.7% of the infants developed blood culture-positive septicemia within five days of life; therefore, this variable was not included in further analyses. In 73.3% and 33.3% of the infants, enteral feeding and probiotics were commenced within five days of life, whereas enteral feeding and probiotics were commenced by the time of blood sampling in 31.8% and 18.0% of the infants, respectively. The serum α1AG levels (n = 255) were assessed on days 2.1 ± 1.8, confirming the early (day 0.3) and late (day 3.9) postnatal ages at blood sampling, for which the regression slopes between independent variables and α1AG levels were determined by the post hoc simple slope analysis.

### 3.2. Crude Dependence of Alpha 1 Acid Glycoprotein Levels on Clinical Variables

A univariable analysis showed that the α1AG levels were positively associated with HDP (*p* < 0.001), gestational age (*p* = 0.039), postnatal age (*p* < 0.001), the commencement of enteral nutrition (*p* = 0.001) and probiotics (*p* < 0.001), the use of antibiotics (*p* = 0.027), intestinal perforation (*p* = 0.001), and CRP levels (*p* < 0.001) and negatively associated with PROM (*p* = 0.002) and body weight z-scores (*p* = 0.010) (Table 2).

### 3.3. Dependence of Alpha 1 Acid Glycoprotein Levels on Postnatal Age, Clinical Variables, and Their Interactions

A significant interaction was observed between postnatal age and HDP (*p* < 0.001), gestational age (*p* = 0.007), cesarean delivery (*p* = 0.018), PROM (*p* = 0.001), chorioamnionitis (*p* < 0.001), funisitis (*p* < 0.001), body weight z-score at birth (*p* < 0.001), the commencement of enteral nutrition (*p* = 0.001), and probiotics (*p* = 0.025) by the time of blood sampling (Table 3).

### 3.4. Independent Variables of Early and Late Alpha 1 Acid Glycoprotein Levels

The post hoc simple slope analysis showed that chorioamnionitis (*p* < 0.001), funisitis (*p* = 0.045), the commencement of enteral feeding (*p* = 0.022), and probiotics (*p* = 0.005) were associated with the early elevation of α1AG levels, whereas caesarean delivery (*p* = 0.025) was associated with an early reduction in the α1AG levels (Table 4 and Online Appendix A). HDP (*p* < 0.001) and gestational age (*p* = 0.001) were associated with a late elevation in α1AG levels, whereas PROM (*p* < 0.001), funisitis (*p* = 0.021), body weight z-score at birth (*p* < 0.001), and the commencement of enteral feeding (*p* = 0.045) were associated with a late reduction in the α1AG levels.

## 4. Discussion

Our findings suggest that the α1AG levels in ELBWIs are related to clinical variables, such as age, growth, intrauterine inflammation, delivery, and postnatal feeding. Chorioamnionitis, funisitis, and the initiation of enteral nutrition and probiotics were associated with an early elevation of α1AG levels. The α1AG values shortly after birth may reflect the immune reaction induced by antenatal exposure to inflammation and postnatal sensitization to extrinsic microbiota. HDP and low z-scores of body weight were also associated with elevated α1AG levels towards the end of the first week of life. It is unclear whether the delayed elevation of α1AG is a direct consequence of the restricted energy substrates in utero or an indirect result of the delay in enteral nutrition, as is often observed in small-for-gestational-age infants [21,22].

### 4.1. Influence of Intrauterine Inflammation and Delivery on Temporal Changes in the α1AG Levels after Birth

Consistent with previous studies on newborn infants, greater gestational and postnatal ages were associated with higher α1AG levels in our extremely preterm cohort. When the effect of clinical variables on the α1AG levels was assessed by incorporating their interaction with postnatal age, the covert roles of clinical conditions suggestive of antenatal inflammation, delivery, and postnatal enteral feeding were highlighted. Chorioamnionitis and funisitis were associated with an early increase in α1AG levels, whereas funisitis and PROM were associated with a late decrease in the α1AG levels. A study on human newborn infants reported that the level of interleukin-8 in cord blood is elevated in association with chorioamnionitis [23], whereas the elevation of interleukin-8 facilitates the production of α1AG [24]. Intrauterine inflammation, represented by chorioamnionitis, funisitis, and PROM, causes a range of adverse events in multiple organs, such as the lung, gut, and immune system [25]. Studies in preclinical models demonstrated that maternal inflammation alters the immune reactions of the fetuses and newborn infants [26,27]. Another study investigated preterm infants born at 23–32 weeks’ gestation and found that the level of monocyte major histocompatibility complex class II expression fell by day 2 from the level of the cord blood and recovered only incompletely by day 7, suggesting the presence of immune paralysis following PROM and preterm labor [28].

Our study also observed that vaginal delivery was associated with an early elevation of the α1AG levels compared to caesarean delivery. This finding was consistent with a previous report involving 70 newborn infants born at gestational weeks 27–42 [13]. Given that the interleukin-6 concentrations in cord blood were higher in infants delivered vaginally than in those delivered via caesarean section and that interleukin-6 accelerates the production of α1AG [9,29,30], it is possible to speculate that, in our current cohort, preterm infants born vaginally experienced exposure to extrinsic microbes earlier than their peers born via caesarean section, leading to elevated serum α1AG levels shortly after birth. The antenatal exposure of preterm infants to in utero inflammation and extrinsic microbes may cause an early elevation of α1AG levels after birth, potentially followed by a suppression phase towards the end of the first week of life. Further investigation of α1AG in preterm infants may help uncover the mechanism of the latent period when infants have already been exposed to microbes, but the biological barrier has not yet been broken down.

### 4.2. Enteral Feeding, Probiotics, and Temporal Changes in the α1AG Levels after Birth

Our study revealed that the early commencement of enteral nutrition and probiotics was associated with elevated α1AG levels shortly after birth. The gut of a newborn infant is sterile at birth but is rapidly colonized by bacteria of maternal and environmental origin [31]. Large amounts of *Lactobacillus* and *Bifidobacterium* are delivered through breast milk, formula, and probiotics, contributing to the establishment of gut microbiota and the maturation of gut immunity [32]. A failure to establish healthy gut microbiota in newborn infants has been linked to an increased risk of morbidity, including hypertension, obesity, and skin and food allergies [33]. Considering that certain types of probiotics facilitate the production of interleukin-6 and that interleukin-6 increases α1AG production [34,35], it is reasonable to speculate that an early commencement of enteral feeding and probiotics led to elevated α1AG levels in the current study. Our study also revealed that an early elevation of α1AG levels in relation to enteral nutrition was followed by a late decrease, the pattern of which mimicked that of funisitis. The level of α1AG might reflect the progress of the homeostatic immune response to the exposure of preterm infants to microbiota via the gut, as well as to pathogenic microbes causing serious systemic infection. However, it is also possible that non-biological substances, such as oligosaccharides in human milk, might have attenuated the production of α1AG. He et al. reported that human milk oligosaccharide supplementation in a cellular model of intestinal epithelial cells decreased the interleukin-8 levels in a time-dependent manner [36]. Given that the early commencement of enteral feeding and probiotics is getting common in preterm infants, it is important to address the precise effect of these practices on immune response in future studies.

### 4.3. Shortage of Energy Substrates in Utero and Temporal Changes in α1AG Levels after Birth

In addition to the clinical conditions indicative of sensitization of the immune system in utero and after birth, our study revealed that HDP and lower z-scores of body weight at birth were associated with a late elevation of α1AG levels. Considering that HDP is one of the major upstream conditions of fetal growth restriction [37] and that the incidence of HDP was indeed associated with lower z-scores of body weight at birth in our study cohort (Online Appendix A), it is reasonable that HDP and lower z-scores similarly contributed to the elevation of α1AG levels. However, as our study cohort comprised ELBWIs, and, therefore, infants with a body weight ≥ 1000 g at birth were excluded, an inverse relationship was inevitably introduced between the z-score of body weight and gestational age, leaving an uncertainty regarding which of HDP, z-scores, and gestational age contributed to an alteration in the α1AG levels. Future studies are needed to investigate whether prolonged placental dysfunction, represented by the presence of HDP and intrauterine growth restriction, alters α1AG production after birth via a non-inflammatory pathway.

### 4.4. Limitations of the Study

Because of the limited population size, the number of infants who developed septicemia and other types of systemic infections was limited, resulting in difficulties in investigating their association with temporal changes in the α1AG levels. Similarly, although we consistently observed postnatal age-specific dependences of α1AG levels on the clinical events suggestive of perinatal inflammation, delivery, and postnatal enteral feeding, related mechanisms were not fully explained because of the lack of studies reporting similar dynamic changes in the immune response after preterm birth. Further studies are needed to confirm the role of these variables in the postnatal changes in the α1AG levels shortly after birth. This was a retrospective observational study based on the patient data collected for clinical reasons. Therefore, only α1AG, CRP, and haptoglobin levels were assessed as inflammatory biomarkers. The assessment of additional biomarkers, such as interleukin-6, interleukin-8, and tumor necrosis factor α, might help to improve the understanding of the relationship between clinical observations and α1AG levels [38,39]. In addition, we were unable to assess the temporal changes in the gut microbiota. Finally, we did not assess the inflammatory biomarkers in the mothers of the participants. Although α1AG is not transferred through the placenta, such information may help in assessing the impact of maternal inflammation on that of the fetus and newborn infant.

## 5. Conclusions

In preterm infants, the incidence of chorioamnionitis and funisitis, vaginal delivery, early enteral feeding, and early probiotics were associated with elevated α1AG levels within the first few days of life, whereas PROM and funisitis were associated with lower α1AG levels towards the end of the first week. Although the mechanisms related with such dynamic postnatal changes in the immune response remain mostly unknown, the intrauterine exposure of fetuses to inflammation and the postnatal sensitization of infants to the extrinsic microbiome may cause the early activation of α1AG production with or without its attenuation thereafter. Future studies are needed to reveal the physiological relevance of the temporal changes in the immune response after birth and to elucidate exactly which stage of sensitization and infection is associated with prominent α1AG elevation. In contrast, HDP, lower z-scores of body weight, and higher gestational age were associated with elevated α1AG levels after a few days of life. Although the dependence of α1AG levels on HDP and body weight z-score can be attributed to the variation in intrauterine growth, these relationships can alternatively be explained solely by gestational age, considering the relationship of HDP (positive) and z-scores (negative) with gestational age. Further studies are required to test whether α1AG is a suitable biomarker for alerts during the latent period of serious systemic infections in vulnerable preterm infants.

## Figures and Tables

**Table 1 nutrients-14-05162-t001:** Clinical backgrounds.

Variables	
Maternal variables	n = 75
Antenatal steroid	39 (52.0%)
Premature rupture of membranes	26 (34.7%)
Hypertensive disorders of pregnancy	18 (24.0%)
Maternal antibiotics	23 (30.7%)
Chorioamnionitis	14 (18.7%)
Funisitis	13 (17.3%)
Variables at birth	n = 75
Male sex	32 (42.7%)
Gestational age in weeks	26.5 ± 2.2
Birth weight in grams	746 ± 162
z-score of birth weight	−1.2 ± 1.3
Apgar score (1 min)	4 {2,6}
Apgar score (5 min)	6 {5,8}
Cesarean delivery	60 (80.0%)
Postnatal variables	n = 75
Postnatal antibiotics	16 (21.3%)
Septicemia	2 (2.7%)
Intestinal perforation	4 (5.3%)
Patent ductus arteriosus requiring surgery	11 (14.7%)
Grade III/IV intraventricular hemorrhage	17 (22.7%)
Postnatal variables at the time of blood sampling	n = 255 *
Postnatal age in days	2.1 ± 1.8
Commencement of enteral nutrition	81 (31.8%)
Commencement of probiotics	46 (18.0%)

Values are shown as the number (%), mean ± standard deviation, or median {quartile ranges}. * represents the number of blood samples.

**Table 2 nutrients-14-05162-t002:** Crude dependence of alpha 1 acid glycoprotein levels on clinical variables.

Variables	Regression Coefficient			*p*
	Mean	95% CI		
		Lower	Upper	
Maternal variables				
Antenatal steroid	1.481	−5.495	8.456	0.677
Premature rupture of membranes	−10.539	−17.263	−3.815	0.002
Hypertensive disorders of pregnancy	11.808	6.210	17.406	<0.001
Maternal antibiotics	1.10	−6.47	8.66	0.776
Chorioamnionitis	8.469	−1.517	18.456	0.096
Funisitis	0.619	−7.862	9.099	0.886
Variables at birth				
Male sex	6.186	−0.817	13.190	0.083
Gestational age in weeks	1.361	0.071	2.650	0.039
Birth weight in grams	0.005	−0.015	0.025	0.632
z-score of birth weight	−2.894	−5.091	−0.698	0.010
Apgar score (1 min)	0.406	−1.018	1.831	0.576
Apgar score (5 min)	0.423	−1.133	1.979	0.594
Cesarean delivery	−6.556	−15.516	2.404	0.152
Postnatal variables				
Postnatal antibiotics	11.386	1.323	21.449	0.027
Septicemia	40.101	11.432	68.770	0.006
Intestinal perforation	17.574	7.246	27.901	0.001
Patent ductus arteriosus requiring surgery	3.351	−5.682	12.385	0.467
Grade Ⅲ/Ⅳ intraventricular hemorrhage	−1.599	−8.550	5.351	0.652
Postnatal variables at blood sampling				
Postnatal age in days	6.689	5.425	7.952	<0.001
Commencement of enteral nutrition	11.395	4.959	17.830	0.001
Commencement of probiotics	16.687	9.023	24.351	<0.001
C-reactive protein (mg/dL)	14.831	10.266	19.396	<0.001

Abbreviation: CI, confidence interval.

**Table 3 nutrients-14-05162-t003:** Dependence of alpha 1 acid glycoprotein levels on postnatal age, clinical variables, and their interactions.

Variables	Regression Coefficient			*p*
	Mean	95% CI		
		Lower	Upper	
Maternal variables				
Antenatal steroid	0.805	−5.267	6.877	0.795
Postnatal age (day)	6.574	4.510	8.639	<0.001
Antenatal glucocorticoid × Postnatal age	−0.035	−2.621	2.550	0.979
PROM	−9.218	−15.220	−3.215	0.003
Postnatal age (day)	7.635	6.088	9.183	<0.001
PROM x Postnatal age	−3.816	−5.982	−1.651	0.001
HDP	7.521	2.721	12.321	0.002
Postnatal age (day)	4.815	3.677	5.952	<0.001
HDP x Postnatal age	4.902	2.541	7.263	<0.001
Maternal antibiotics	0.799	−5.837	7.436	0.813
Postnatal age (day)	6.991	5.639	8.342	<0.001
Maternal antibiotics x Postnatal age	−1.290	−4.211	1.630	0.386
Chorioamnionitis	8.639	−0.837	18.116	0.074
Postnatal age (day)	7.568	6.209	8.928	<0.001
Chorioamnionitis x Postnatal age	−5.089	−7.480	−2.698	<0.001
Funisitis	0.387	−7.841	8.616	0.927
Postnatal age (day)	7.619	6.290	8.948	<0.001
Funisitis x Postnatal age	−5.227	−7.749	−2.705	<0.001
Variables at birth				
Male sex	5.968	−0.187	12.124	0.057
Postnatal age (day)	7.201	5.576	8.826	<0.001
Male sex x Postnatal age	−1.544	−4.121	1.033	0.24
Gestational age (week)	1.202	0.056	2.348	0.04
Postnatal age (day)	6.585	5.382	7.787	<0.001
Gestational age x Postnatal age	0.607	0.165	1.050	0.007
Birth weight	0.008	−0.008	0.025	0.314
Postnatal age (day)	6.534	5.314	7.749	<0.001
Birth weight x Postnatal age	−0.005	−0.013	0.003	0.198
z-score of birth weight	−2.114	−4.029	−0.199	0.031
Postnatal age (day)	6.392	5.401	7.382	<0.001
z-score of birth weight x Postnatal age	−2.002	−2.770	−1.235	<0.001
Apgar score (5 min)	0.337	−0.995	1.670	0.62
Postnatal age (day)	6.597	5.336	7.857	<0.001
Apgar score (5 min) x Postnatal age	0.175	−0.296	0.645	0.466
Cesarean delivery	−6.415	−14.998	2.169	0.143
Postnatal age (day)	3.918	1.577	6.258	0.001
Cesarean delivery x Postnatal age	3.316	0.575	6.056	0.018
Postnatal variables				
Postnatal antibiotics	6.625	−1.065	14.315	0.091
Postnatal age (day)	5.755	4.515	6.995	<0.001
Postnatal antibiotics x Postnatal age	2.718	−0.205	5.641	0.068
Septicemia	88.338	18.734	157.942	0.013
Postnatal age (day)	6.364	5.177	7.550	<0.001
Septicemia x Postnatal age	−24.364	−58.931	10.204	0.167
Intestinal perforation	4.592	−2.055	11.238	0.176
Postnatal age (day)	6.401	5.061	7.740	<0.001
Intestinal perforation x Postnatal age	1.433	−2.126	4.991	0.43
PDA requiring surgery	2.891	−5.767	11.550	0.513
Postnatal age (day)	6.859	5.454	8.263	<0.001
PDA requiring surgery x Postnatal age	−2.001	−5.077	1.076	0.202
Severe IVH *	−1.920	−7.812	3.972	0.523
Postnatal age (day)	6.842	5.315	8.369	<0.001
Severe IVH x Postnatal age	−1.257	−3.568	1.054	0.286
Postnatal variables at blood sampling				
Enteral nutrition **	3.375	−4.588	11.338	0.406
Postnatal age (day)	8.022	6.498	9.545	<0.001
Enteral nutrition ** x Postnatal age	−5.548	−8.777	−2.319	0.001
Probiotics **	12.354	3.136	21.573	0.009
Postnatal age (day)	6.677	5.248	8.106	<0.001
Probiotics ** x Postnatal age	−4.720	−8.859	−0.580	0.025
C-reactive protein (mg/dL)	14.553	10.083	19.023	<0.001
Postnatal age (day)	6.631	5.476	7.786	<0.001
C-reactive protein x Postnatal age	−0.938	−5.391	3.515	0.68

* Defined as grade III/IV IVH. ** Commenced by the time of blood sampling. Abbreviations: CI, confidence interval; HDP, hypertensive disorders of pregnancy; IVH, intraventricular hemorrhage; PDA, patent ductus arteriosus; PROM, premature rupture of membranes.

**Table 4 nutrients-14-05162-t004:** Independent variables of alpha 1 acid glycoprotein levels during early and late phases.

Variables	Early *				Late *			
	Regression Coefficient			*p*	Regression Coefficient			*p*
	Mean	95% CI			Mean	95% CI		
		Lower	Upper			Lower	Upper	
Premature rupture of membranes	−1.585	−7.999	4.829	0.628	−16.850	−25.122	−8.578	<0.001
Hypertensive disorders of pregnancy	−2.282	−8.317	3.752	0.459	17.324	9.958	21.249	<0.001
Chorioamnionitis	18.817	8.500	29.133	<0.001	−1.538	−12.442	9.366	0.782
Funisitis	10.841	0.220	21.463	0.045	−10.067	−18.639	−1.494	0.021
Gestational age in weeks	−0.013	−1.457	1.431	0.986	2.416	0.965	3.868	0.001
z-score of birth weight	1.891	−0.389	4.171	0.104	−6.119	−8.736	−3.502	<0.001
Cesarean delivery	−13.046	−24.449	−1.643	0.025	0.217	−8.582	9.015	0.961
Commencement of enteral nutrition	14.471	2.096	26.846	0.022	−7.721	−15.276	−0.167	0.045
Commencement of probiotics	21.794	6.724	36.863	0.005	2.915	−6.028	11.858	0.523

* Simple slopes were calculated for one standard deviation below (Early, 0.3) and above (Late, 3.9) the mean postnatal age. Abbreviation: CI, confidence interval.

## Data Availability

The data presented in this study are available on request from the corresponding author.

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
