# Peer review of "Influence of Intrauterine Inflammation, Delivery, and Postnatal Feeding on the Temporal Changes of Serum Alpha 1 Acid Glycoprotein Levels in Extremely-Low-Birth-Weight Infants"

_nutrients, 2022, doi:10.3390/nu14235162_

Round 1

Reviewer 1 Report

Influence of intrauterine inflammation, delivery, and postnatal feeding on temporal changof serum alpha 1 acid glycoprotein level in extremely low birth weight infants

Yasuhisa Nakamura et al.

Infection in neonatal period of life is the major cause of mortality. In order to obtain an early diagnosis of infection, a number of blood tests has been proposed including alpha 1 acid glycoprotein (a1AG), an acute-phase protein. In order to study the mechanisms responsible of an increased level of serum a1AG, a retrospective study has been carried out in a population of 75 preterm newborns weighing less than 1,000 g. Significantly Increased levels of a1AG were found in infants born from mothers with premature rupture of membranes or hypertensive disorders of pregnancy. At birth gestational age and z-score of birth weight were the more important influencing factors while after birth septicemia, antibiotic administration, and intestinal perforation were significantly important. Postnatal age, enteral nutrition, and probiotic administration were connected with high levels of  a1AG.  

These data are interesting because they raise serious doubts on the efficacy of a1AG serum determination in finding initial serious infections in preterm newborns. On the other hand, the effect of probiotics on the level of alpha 1 acid glycoprotein must be further studied, and, in my opinion, deserves a few words of comment.

Author Response

We thank the reviewer for the precise summary of and positive comments on our study. The Discussion section has been revised to emphasize the importance of investigating the impact of probiotics to α1AG levels.

Reviewer 2 Report

Nakamura and colleagues present a descriptive study of serum alpha 1 acid glycoprotein (α1AG) levels in extremely low birth weight infants in association with numerous clinical outcomes.

In general, the statistical analyses could be better described. What statistical software was used to conduct these analyses? Were attempts made to use a multivariate framework incorporating all of the significant clinical variables to model α1AG levels?

With respect to clinical backgrounds (i.e., table 1) no information is presented on the levels of α1AG, CRP, or haptoglobin at early or later postnatal ages. Can this be reported and/or summarized?

The main text in sections 3.2, 3.3, and 3.4 are redundant with the information provided in tables 2, 3, and 4, respectively. The presentation of the results in the tables is more readily understood. Perhaps these sections could be rewritten to more broadly summarize how many variables were significant and then limit the text to describe only the variables with the largest coefficients in both directions and point the reader to the tables for the full results.  

Table 4 and discussion - A more detailed discussion providing some additional context or thoughts/hypotheses on why the coefficients are reversed from early to late phase, e.g., funisitis and enteral nutrition, would add to the depth of the paper.

Minor comments:

Ln 59, Typo? Should this be "breach of biological barriers"?

Ln 204 – Causal language should be removed.

Ln 250 – Genera names should be italicized.

Author Response

Reviewer Comment:

Nakamura and colleagues present a descriptive study of serum alpha 1 acid glycoprotein (α1AG) levels in extremely low birth weight infants in association with numerous clinical outcomes.

In general, the statistical analyses could be better described. What statistical software was used to conduct these analyses? Were attempts made to use a multivariate framework incorporating all of the significant clinical variables to model α1AG levels?

  • Response to the reviewer: We accept the reviewer’s comment that the original Methods section was too brief for readers to figure out the exact steps of data analysis. We have provided the detailed information of the software in the revised Methods section. The flow of the analysis was, Step 1: Univariate analysis to assess the dependence of α1AG levels on clinical variables without adjusting for covariates, Step 2: Univariate analysis to assess the dependence of α1AG levels on the clinical variables, postnatal age, and their interactions, and Step 3: Post hoc simple slope analysis to visualize the slope of regressions between the clinical variables and α1AG levels at representative postnatal ages. We did not perform a multivariable analysis explaining the relationship between clinical variables and α1AG levels, because this was not the main scope of our study, and also because the clinical variables used in our analysis were tightly inter-correlated between each other, rendering these variables unsuitable to use within the same model. We have revised the Data Analysis in the Methods section to provide detailed information regarding the statistical analysis used. We thank the reviewer for the comment.

Reviewer Comment:

With respect to clinical backgrounds (i.e., table 1) no information is presented on the levels of α1AG, CRP, or haptoglobin at early or later postnatal ages. Can this be reported and/or summarized?

  • Response to the reviewer: The post hoc simple slope analysis allows the estimation of the slope of regression lines at two postnatal ages, or early (one standard deviation below the mean postnatal age, or 0.3 days in our current data) and late (one standard deviation above the mean postnatal age, or 3.9 days in our data). This analysis is powered to estimate the “slope” of regression between variables by incorporating the influence of covariates (e.g. postnatal age in our current study), but not to give the estimation of dependent variables (e.g.α1AG in our study) at these time points. Having described these, the authors understand the reviewer’s and readers’ interest in the α1AG levels related with the different slopes of regression. We have worked out with our biostatistician to present a new Online Supplementary Figure depicting the slope and approximate level of α1AG for the early and late postnatal ages, and to revise the Methods section for clarity. The authors hope that the revise Methods section is now clear enough to make readers understood with the detailed analytical approach.

Reviewer Comment:

The main text in sections 3.2, 3.3, and 3.4 are redundant with the information provided in tables 2, 3, and 4, respectively. The presentation of the results in the tables is more readily understood. Perhaps these sections could be rewritten to more broadly summarize how many variables were significant and then limit the text to describe only the variables with the largest coefficients in both directions and point the reader to the tables for the full results.

  • Response to the reviewer: We agree that the tables are informative and self-explanatory, and therefore, it is redundant to show all of the correlation coefficients, their confidence intervals, and p-values in the text. Although recent recommendations in core journals, such in NEJM, are to show confidence intervals rather than p-values, we decided to simply show the p-values after discussing with our biostatistician. We thank the reviewer for the suggestion. Regarding the presentation of representative variables in the text, we consider this strategy inappropriate, because, as explained in the response to the previous query, we did not perform multivariable analysis based on the findings of univariable analysis. The authors appreciate the reviewer’s understanding.

Reviewer Comment:

Table 4 and discussion - A more detailed discussion providing some additional context or thoughts/hypotheses on why the coefficients are reversed from early to late phase, e.g., funisitis and enteral nutrition, would add to the depth of the paper.

  • Response to the reviewer: We agree that the readers may be interested in the mechanism of the temporally paradoxical relationship between antenatal inflammation, enteral feeding andα1AG levels. However, thus far, we have been unable to identify articles reporting similar findings to ours i.e. a temporal conversion from immune activation to possible immune paralysis in newborn infants, except for those related with severe systemic infection, such as septicemia. We have emphasized in the Limitation section that further studies are needed addressing the dynamic change in immune response following a range of clinical events. We hope the reviewer understands the current situation.

Reviewer Comment:

Minor comments:

Ln 59, Typo? Should this be "breach of biological barriers"?

  • Response to the reviewer: The authors appreciate the reviewer’s suggestion. The text has been revised as suggested.

Reviewer Comment:

Ln 204 – Causal language should be removed.

  • Response to the reviewer: The authors are grateful to the reviewer for the comment. We accept that our findings do not suggest a causal relationship between variables, and, therefore, the term “determined” has been replaced by “related” in the revised Discussion section.

Reviewer Comment:

Ln 250 – Genera names should be italicized.

  • Response to the reviewer: Lactobacillus and Bifidobacterium have been italicized. We thank the reviewer for the suggestion.